# Recurrent Neural Networks Learn to Store and Generate Sequences using Non-Linear Representations

## Abstract

The Linear Representation Hypothesis (LRH) states that neural networks learn to encode concepts as directions in activation space, and a strong version of the LRH states that models learn *only* such encodings. In this paper, we present a counterexample to this strong LRH: when trained to repeat an input token sequence, gated recurrent neural networks (RNNs) learn to represent the token at each position with a particular order of magnitude, rather than a direction. These representations have layered features that are impossible to locate in distinct linear subspaces. To show this, we train interventions to predict and manipulate tokens by learning the scaling factor corresponding to each sequence position. These interventions indicate that the smallest RNNs find only this magnitude-based solution, while larger RNNs have linear representations. These findings strongly indicate that interpretability research should not be confined by the LRH.

## 1 Introduction

It has long been observed that neural networks encode concepts as linear directions in their representations (Smolensky, 1986), and much recent work has articulated and explored this insight as the Linear Representation Hypothesis (LRH; Elhage et al. 2022; Park et al. 2023; Guerner et al. 2023; Nanda et al. 2023; Olah 2024). A *strong* interpretation of the LRH says that such linear encodings are entirely sufficient for a mechanistic analysis of a deep learning model (Smith, 2024).

In this paper, we present a counterexample to the Strong LRH by showing that recurrent neural networks with Gated Recurrent Units (GRUs; Cho et al. 2014) learn to represent the token at each position using magnitude rather than direction when solving a simple repeat task (memorizing and generating a sequence of tokens). This leads to a set of layered features that are impossible to locate in dis-

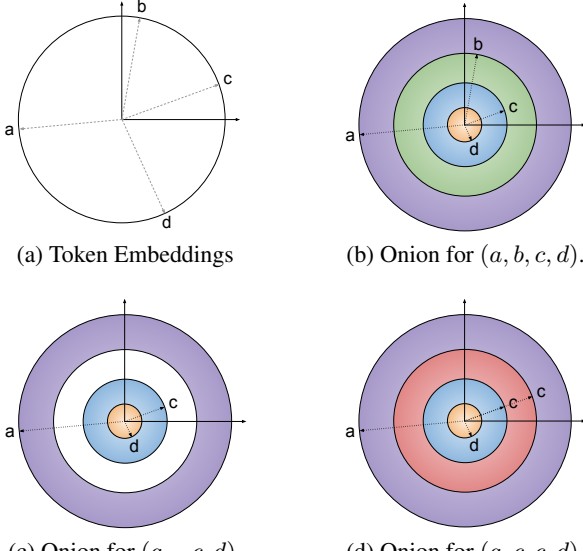

(a) Token Embeddings  (b) Onion for $(a, b, c, d)$.

(c) Onion for $(a, \cdot, c, d)$.  (d) Onion for $(a, c, c, d)$.

Figure 1: We find that GRUs solve a repeat task by learning a scaling factor corresponding to each sequence position, leading to layered onion-like representations. In this simplified illustration, the learned token embeddings (a) are rescaled to have magnitudes proportional to their sequence positions (b). To change an element of the sequence, remove (c) and replace (d) the token embedding at the given positional magnitude. The layered nature of the representations makes them non-linear; any direction will cross-cut multiple layers of the onion.

tinct linear subspaces. We refer to the resulting hidden states as 'onion representations' to evoke how sequence position can be identified by iteratively peeling off these magnitude changes from the positions before it (Figure 1). In our experiments, this is the only solution found by the smallest networks (hidden size 48, 64); the larger networks (128, 512, 1024) learn to store input tokens in position-specific linear subspaces, consistent with the LRH, though we find these linear representations are compatible with onion-based mechanisms as well.

We made this surprising finding in a hypothesis-driven fashion. Our Hypothesis 1 was that GRUs would store each token in a linear subspace. To

test this hypothesis, we employed a variant of distributed alignment search (DAS; Geiger et al. 2024b; Wu et al. 2023) that uses a Gumbel softmax to select dimensions for intervention. This revealed that the larger GRUs do in fact have linear subspaces for each position, but we found no evidence of this for the smaller ones (section 5). This led to Hypothesis 2: GRUs learn to represent input *bigrams* in linear subspaces. A DAS-based analysis supports this for the medium-sized models but not for the smallest ones (section 6). This left the task success of the smallest models to be explained.

For the smallest models, we observed that the update gates of the GRUs got gradually lower as the sequence progressed. This led to Hypothesis 3: onion representations. To evaluate this hypothesis, we learned interventions on the hidden vector encoding a sequence of tokens that replaces token $A$ with token $B$ at position $j$. The intervention adds the scaled difference of learned embeddings for $A$ and $B$, where the scaling factor is determined by the position $j$ with learned linear and exponential terms. Across positions, this intervention works with ≈90% accuracy, demonstrating the existence of layered features stored at different scales.

The existence of non-linear representations is a well-formed theoretical possibility. For example, under the framework of Geiger et al. (2024a) and Huang et al. (2024), any bijective function can be used to featurize a hidden vector, and interventions can be performed on these potentially non-linear features. However, the typical causal analysis of a neural networks involves only interventions on linear representations (see Section 2 for a brief review of such methods). We hope that our counterexample to the strong version of the LRH spurs researchers to consider methods that fall outside of this class, so that we do not overlook concepts and mechanisms that our models have learned.

## 2 Related Work

**The Linear Representation Hypothesis** Much early work on 'word vectors' was guided by the idea that linear operations on vectors could identify meaningful structure (Mikolov et al., 2013; Arora et al., 2016; Levy and Goldberg, 2014). More recently, Elhage et al. (2022) articulated the Linear Representation Hypothesis (LRH), which says that (1) features are represented as directions in vector space and (2) features are one-dimensional (see also Elhage et al. 2022; Park et al. 2023; Guerner

et al. 2023; Nanda et al. 2023). Engels et al. 2024 challenged (2) by showing some features are irreducibly multi-dimensional. Olah (2024) subsequently argued that (1) is the more significant aspect of the hypothesis, and it is the one that we focus on here. Smith (2024) adds important nuance to the LRH by distinguishing a weak version (some concepts are linearly encoded) from a strong one (all concepts are linearly encoded).

Our concern is with the strong form; there is ample evidence that linear encoding is possible, but our example shows that other encodings are possible. In onion representations, multiple concepts can be represented in a linear subspace by storing each concept at a different order of magnitude, i.e., a 'layer' of the onion, and any direction will cross-cut multiple layers of the onion.

**Intervention-based Methods** Recent years have seen an outpouring of new methods in which interventions are performed on linear representations, e.g., entire vectors (Vig et al., 2020; Geiger et al., 2020; Finlayson et al., 2021; Wang et al., 2023), individual dimensions of weights (Csordás et al., 2021) and hidden vectors (Giulianelli et al., 2018; De Cao et al., 2020; Davies et al., 2023), linear subspaces (Ravfogel et al., 2020; Geiger et al., 2024b; Belrose et al., 2023), or linear features from a sparse dictionary (Marks et al., 2024; Makelov et al., 2024). These methods have provided deep insights into how neural networks operate. However, the vast and varied space of non-linear representations is woefully underexplored in a causal setting.

**RNNs** Recurrent Neural Networks (RNNs) were among the first neural architectures used to process sequential data (Elman, 1990, 1991). Many variants arose to help networks successfully store and manage information across long sequences, including LSTMs (Hochreiter and Schmidhuber, 1997) and GRUs (Cho et al., 2014). Bidirectional LSTMs provided the basis for one of the first large-scale pretraining efforts (ELMo; Peters et al. 2018). With the rise of Transformer-based models (Vaswani et al., 2017), RNNs fell out of favor somewhat, but the arrival of structured state-space models (Gu et al., 2021b,a; Gu and Dao, 2023; Dao and Gu, 2024) has brought RNNs back into the spotlight, since such models seek to replace the Transformer's potentially costly attention mechanisms with recurrent connections. We chose GRUs for our studies, with an eye towards better understanding structured state space models as well.

| | $N = 48$ | $N = 64$ | $N = 128$ | $N = 256$ | $N = 512$ | $N = 1024$ |
|---|---|---|---|---|---|---|
| Exact-Match Accuracy | $0.95 \pm 0.01$ | $0.97 \pm 0.00$ | $1.00 \pm 0.00$ | $1.00 \pm 0.00$ | $1.00 \pm 0.00$ | $1.00 \pm 0.00$ |

Table 1: Exact-match accuracy (mean of 5 runs; $\pm$ 1 s.d.) for GRUs of different sizes trained on the repeat task.

## 3 Models

In this paper, we focus on how RNNs solve the repeat task. As noted in section 2, this question has taken on renewed importance with the development of structured state-space models that depend on recurrent computations and are meant to provide efficient alternatives to transformers.

Define an RNN as $\boldsymbol{h}_t = f(\boldsymbol{h}_{t-1}, \boldsymbol{x}_t)$, $\boldsymbol{h}_0 = 0$, where $f(\cdot, \cdot)$ is the state update function, $t \in \{1, \ldots, T\}$ is the current timestep, $\boldsymbol{x}_t \in \mathbb{R}^N$ is the current input, and $\boldsymbol{h}_t \in \mathbb{R}^N$ is the state after receiving the input $\boldsymbol{x}_t$. The output of the model is $\boldsymbol{y}_t = g(\boldsymbol{h}_t)$. Vectorized inputs $\boldsymbol{x}_t$ are obtained with a learned embedding $\boldsymbol{E} \in \mathbb{R}^{N_S \times N}$, using the indexing operator $\boldsymbol{x}_t = \boldsymbol{E}[i_t]$, where $i_t \in \{1, \ldots, N_S\}$ is the index of the token at timestep $t$.

In our experiments, we use GRU cells over the more widely-used LSTM cells because they have a single state to intervene on, as opposed to the two states of the LSTM. GRU-based RNNs defined as:

$$\boldsymbol{z}_t = \sigma \left( \boldsymbol{W}_z \boldsymbol{x}_t + \boldsymbol{U}_z \boldsymbol{h}_t + \boldsymbol{b}_z \right) \tag{1}$$

$$\boldsymbol{r}_t = \sigma \left( \boldsymbol{W}_r \boldsymbol{x}_t + \boldsymbol{U}_r \boldsymbol{h}_t + \boldsymbol{b}_r \right) \tag{2}$$

$$\boldsymbol{u}_t = \tanh \left( \boldsymbol{W}_h \boldsymbol{x}_t + \boldsymbol{U}_h (\boldsymbol{r}_t \odot \boldsymbol{h}_t) + \boldsymbol{b}_h \right) \tag{3}$$

$$\boldsymbol{h}_t = (1 - \boldsymbol{z}_t) \odot \boldsymbol{h}_{t-1} + \boldsymbol{z}_t \odot \boldsymbol{u}_t \tag{4}$$

For output generation, we use $g(\boldsymbol{h}_t) = \mathrm{softmax}(\boldsymbol{h}_t \boldsymbol{W}_o + \boldsymbol{b}_o)$. The learned parameters are weights $\boldsymbol{W}_*, \boldsymbol{U}_* \in \mathbb{R}^{N \times N}$, and biases $\boldsymbol{b}_* \in \mathbb{R}^N$.

We will investigate how the final hidden state $\boldsymbol{h}_L$ of a GRU represents an input token sequence $\mathbf{i} = i_1, i_2, \ldots i_L$. The final state is a bottle-neck between the input token sequence and the output.

## 4 Repeat Task Experiments

Our over-arching research question is how different models learn to represent abstract concepts. The repeat task is an appealingly simple setting in which to explore this question. In this task, the network is presented with a sequence of random tokens $\mathbf{i} = i_1, i_2, \ldots, i_L$, where each $i_j$ is chosen with replacement from a set of symbols $N_S$ and the length $L$ is chosen at random from $\{1 \ldots L_{\max}\}$. This is followed by a special token, $i_{L+1} = $ 'S', that indicates the start of the repeat phase. The task is

to repeat the input sequence: $y_{L+1+j} = i_j$. The variables in this task will represent positions in the sequence and take on token values.

As a preliminary step, we evaluate RNN models on the repeat task. The core finding is that all of the models solve the task. This sets us up to explore our core interpretability hypotheses in sections 5–7.

### 4.1 Setup

For our experiments, we generate 1M random sequences of the repeat task. The maximum sequence length is $L_{\max} = 9$, and the number of possible symbols is $N_S = 30$. For testing, we generate an additional 5K examples using the same procedure, ensuring that they are disjoint at the sequence level from those included in the train set.

We use the same model weights during both the input and decoding phases. During the input phase, we ignore the model's outputs. No loss is applied to these positions. We use an autoregressive decoding phase: the model receives its previous output as input in the next step. We investigate multiple hidden state sizes, from $N = 48$ to $N = 1024$.

We train using a batch size of 256, up to 40K iterations, which is sufficient for each model variants to converge. We use an AdamW optimizer with a learning rate of $10^{-3}$ and a weight decay of 0.1.

### 4.2 Results

Table 1 reports on model performance at solving the repeat task. It seems fair to say that all the models solve the task; only the smallest model comes in shy of a perfect score, but it is at 95%. Overall, these results provide a solid basis for asking *how* the models manage to do this. This is the question we take up for the remainder of the paper.

## 5 Hypothesis 1: Unigram Variables

Intuitively, to solve the repeat task, the token at each position will have a different feature in the state vector $\boldsymbol{h}_L$ (the boundary between the input and output phrases). In line with the LRH, we hypothesize these features will be linear subspaces.

| Intervention | $N = 48$ | $N = 64$ | $N = 128$ | $N = 256$ | $N = 512$ | $N = 1024$ |
|---|---|---|---|---|---|---|
| Linear Unigram | $0.00 \pm 0.00$ | $0.00 \pm 0.00$ | $0.01 \pm 0.00$ | $0.18 \pm 0.03$ | $0.91 \pm 0.08$ | $1.00 \pm 0.00$ |
| Linear Bigram | $0.01 \pm 0.00$ | $0.01 \pm 0.00$ | $0.54 \pm 0.05$ | $0.97 \pm 0.05$ | $1.00 \pm 0.00$ | $1.00 \pm 0.00$ |
| Onion Unigram | $0.83 \pm 0.03$ | $0.87 \pm 0.03$ | $0.89 \pm 0.04$ | $0.91 \pm 0.08$ | $0.95 \pm 0.01$ | $0.94 \pm 0.04$ |

Table 2: Intervention accuracy (mean of 5 runs; $\pm$ 1 s.d.) for GRUs of different sizes trained on the repeat task.

## 5.1 Interchange Intervention Data

In causal abstraction analysis (Geiger et al., 2021), interchange interventions are used to determine the content of a representation by fixing it to the counterfactual value it would have taken on if a different input were provided. These operations require datasets of counterfactuals. To create such examples, we begin with a random sequence $\mathbf{y}$ of length $L$ consisting of elements of our vocabulary. We then sample a set of positions $I \subseteq \{1, \ldots, L\}$, where each position $k$ has a 50% chance of being selected. To create the base $\mathbf{b}$, we copy $\mathbf{y}$ and then replace each $b_k$ with a random token, for $k \in I$. To create the source $\mathbf{s}$, we copy $\mathbf{y}$ and then replace each $s_j$ with a random token, for $j \notin I$. Here is a simple example with $I = \{1, 3\}$:

$$\mathbf{y} = \text{b d a c}$$
$$\mathbf{b} = \text{X d Y c}$$
$$\mathbf{s} = \text{b 4 a 1}$$

Our core question is whether we can replace representations obtained from processing $\mathbf{b}$ with those obtained from processing $\mathbf{s}$ in a way that leads the model to predict $\mathbf{y}$ in the decoding phase.

## 5.2 Method: Interchange Interventions on Unigram Subspaces

Our goal is to localize each position $k$ in the input token sequence to a separate linear subspaces $S_k$ of $h_L$. We will evaluate our success using interchange interventions. For each position in $k \in I$, we replace the subspace $S_k$ in the hidden representation $h_L^{\mathbf{b}}$ for base input sequence $\mathbf{b}$ with the value it takes in $h_L^{\mathbf{s}}$ for source input sequence $\mathbf{s}$. The resulting output sequence should exactly match $\mathbf{y}$. If we succeed, we have shown that the network has linear representations for each position in a sequence.

There is no reason to assume that the subspaces will be axis-aligned. Thus, we use Distributed Alignment Search (DAS) and train a rotation matrix $\boldsymbol{R} \in \mathbb{R}^{N \times N}$ to map $\boldsymbol{h}$ into a new rotated space $\bar{\boldsymbol{h}}$. However, a remaining difficulty is to determine which dimensions in the rotated space belong to which position. The size of individual subspaces may differ: for example, the first input of a repeated sequence, $b_1$, is always present, and the probability of successive inputs decreases due to the random length of the input sequences. Thus, the network might decide to allocate a larger subspace to the more important variables that are always present, maximizing the probability of correct decoding for popular sequence elements.

To solve this problem, we learn an assignment matrix $\boldsymbol{A} \in \{0, 1\}^{N \times (L+1)}$ that assigns dimensions of the axis-aligned representation $\bar{\boldsymbol{h}}$ with at most one sequence position. Allowing some dimensions to be unassigned provides the possibility for the network to store other information that is outside of these positions, such as the input length.

We can learn this assignment matrix by defining a soft version of it $\hat{\boldsymbol{A}} \in \mathbb{R}^{N \times (L+1)}$, and taking the hard gumbel-softmax (Jang et al., 2017; Maddison et al., 2017) with straight-through estimator (Hinton, 2012; Bengio et al., 2013) over its columns for each row ($r \in \{1 \ldots N\}$) independently:

$$\boldsymbol{A}[r] = \text{gumbel\_softmax}(\hat{\boldsymbol{A}}[r]) \qquad (5)$$

For intervening on the position $k \in \mathbb{N}$, we replace dimensions of the rotated state $\bar{\boldsymbol{h}}$, that are 1 in $\boldsymbol{A}[\cdot, v]$. Specifically, intervention $\hat{h}^{\mathbf{b}}$ is defined:

$$\bar{h}^{\mathbf{b}} = \boldsymbol{R}h^{\mathbf{b}} \qquad (6)$$
$$\bar{h}^{\mathbf{s}} = \boldsymbol{R}h^{\mathbf{s}} \qquad (7)$$
$$\hat{\bar{h}}^{\mathbf{b}} = \boldsymbol{A}[\cdot, v] \odot \bar{h}^{\mathbf{s}} + (1 - \boldsymbol{A}[\cdot, v]) \odot \bar{h}^{\mathbf{b}} \qquad (8)$$
$$\hat{h}^{\mathbf{b}} = \boldsymbol{R}^{\mathsf{T}}\hat{\bar{h}}^{\mathbf{b}} \qquad (9)$$

When learning the rotation matrix $\boldsymbol{R}$ and assignment matrix $\boldsymbol{A}$, we freeze the parameters of the already trained GRU network. We perform the intervention on the final state of the GRU, after encoding the input sequences, and use the original GRU to decode the output sequence $\hat{\mathbf{y}}$ from the intervened state $\hat{h}_L^{\mathbf{b}}$. We update $\boldsymbol{R}$ and $\boldsymbol{A}$ by backpropagating with respect to the cross entropy loss between the output sequence $\hat{\mathbf{y}}$ and the expected output sequence after intervention $\mathbf{y}$.

### 5.3 Results

We use the same training set as the base model to train the intervention model, and we use the same validation set to evaluate it. The first row of Table 2 shows the accuracy of the unigram intervention. It works well for "big" models, with $N \geq 512$. In these cases, we can confidentially conclude that the model has a separate linear subspace for each position in the sequence.

### 5.4 Discussion

The above results suggest that the model prefers to store each input element in a different subspace if there is "enough space" in its representations relative to the task. However, Hypothesis 1 seems to be incorrect for autoregressive decoders where $N < 512$. Since these models do solve our task, we need to find an alternative explanation for how they succeed. This leads us to Hypothesis 2.

## 6 Hypothesis 2: Bigram Variables

Our second hypothesis is a minor variant of Hypothesis 1. Here, we posit that, instead of representing variables for unigrams, the model instead stores tuples of inputs $(i_t, i_{t+1})$ we call bigram variables.

### 6.1 Intervention Data

We create counterfactual pairs using the same method as we used for Hypothesis 1 (section 5.1). In this case, each token $i_t$ affects two bigram variables (if present). Thus, the subspace replacement intervention must be performed on both of these variables. This also means that, for each $k \in I$, the tokens $s_{k-1}$ and $s_{k+1}$ in the source sequence input must match $b_{t-1}$ and $b_{t+1}$ in the base sequence, because the bigram at position $t - 1$ depends on $(i_{t-1}, i_t)$ and the bigram at $t$ depends on $(i_t, i_{t+1})$.

### 6.2 Method: Interchange Interventions on Bigram Subspaces

For a sequence of length $L$, there are $L - 1$ bigram variables. To try to identify these, we use the same interchange intervention method described in section 5.2. Because targeting a single position in the base input sequence requires replacing two bigram variables, we intervene on only a single token at a time. Otherwise, the randomized sequence could be too close to the original, and most of the subspaces would be replaced at once, thereby artificially simplifying the task.

### 6.3 Results

We show the effectiveness of bigram interventions in the middle row of Table 2. The intervention is successful on most sizes, but fails for the smallest models ($N \leq 64$).

### 6.4 Discussion

We hypothesize that the models prefer to learn bigram representations because of their benefits for autoregressive input: the current input can be compared to each of the stored tuples, and the output can be generated from the second element of the tuple. This alone would be enough to repeat all sequences which have no repeated tokens. Because our models solve the task with repeat tokens, an additional mechanism must be involved. Regardless, bigrams could provide a powerful representation that is advantageous for the model.

Two additional remarks are in order. First, successful unigram interventions entail successful bigram interventions; a full argument is given in Appendix E.1. Second, one might worry that our negative results for smaller models trace to limitations of DAS on the small models. Appendix E.2 addresses this by showing DAS succeeding on a non-autoregressive control model ($N \leq 64$) that solves the copy task. This alleviates the concern, suggesting that the small autoregressive model does not implement the bigram solution and highlighting the role of autoregression in the bigram solution.

However, we still do not have an explanation for how the smallest models ($N \leq 64$) manages to solve the repeat task; Hypotheses 1 and 2 are unsupported as explanations for this model. This in turn leads us to Hypothesis 3.

## 7 Hypothesis 3: Onion Representations

In an effort to better understand how the smallest GRUs solve the repeat task, we inspected the gate values $z_t$ as defined in equation 1 from the GRU definition (section 3).

Figure 2a visualizes the first 64 input gates for the $N = 1024$ model (Appendix figure 5 is a larger diagram with all the gates). The x-axis is the sequence (temporal dimension) and the y-axis depicts the gate for each dimension. One can see that this model uses gates to store inputs by closing position-dependent channels sharply, creating a position-dependent subspace for each input. (This gating pattern is consistent across all inputs.)

Figure 2b shows all the gates for the $N = 64$

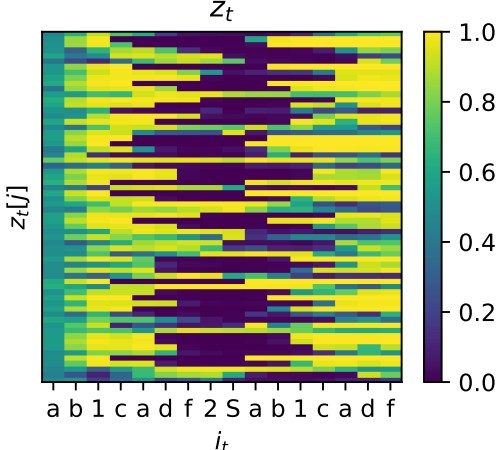

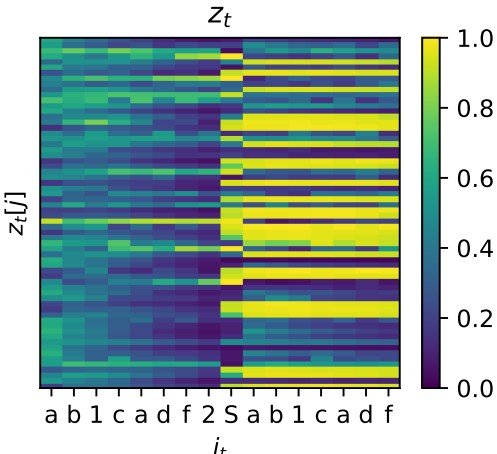

(a) The first 64 channels of GRU with $N = 1024$. The model learns to store variables in different, axis-aligned subspaces. Gates close sharply, freezing individual subspaces at different times. For all channels, please refer to Figure 5 in the Appendix.

(b) GRU with $N = 64$ learns a "onion representation", using different scales of the same numbers to represent the variables. The gates close gradually and synchronously in the input phase, providing the exponentially decaying scaling needed to represent different positions in the sequence.

Figure 2: The input gate $z_t$ in GRUs learning different representations Yellow is open; dark blue is closed; $y$-axis is the channel; $x$ axis is the position. Both models use input gates to let in different proportions of each dimension across the sequence in order to store the positions of the input tokens. The large model (left) sharply turns off individual channels to mark position; in contrast, the small model (right) gradually turns off all channels.

model. Here, the picture looks substantially different. This model gradually closes its gates simultaneously, suggesting that the network might be using this gate to encode token positions. This led us to Hypothesis 3: RNNs learn to encode each position in a sequence as a magnitude.

This hypothesis relies heavily on the autoregressive nature of the GRU, the discriminative capacity of the output classifier $g(\boldsymbol{h}_t)$, and the sequential nature of the problem. Multiple features can be stored in the same subspace, at different scales. When the GRU begins to generate tokens at timestep $t = L + 2$, if the scales $s_{t'}$ associated with position $t' > t$ are sufficiently small ($s_{t'} \ll s_t$), the output classifier $y_t = g(\boldsymbol{h}_t)$ will be able to correctly decode the first input token $i_1$. In the following step, $i_1$ is fed back to the model as an input, and the model is able to remove the scaled representation corresponding to $i_1$ from $\boldsymbol{h}_t$, obtaining $\boldsymbol{h}_{t+1}$. In this new representation, the input with the next largest scale, $i_2$, will be dominant and will be decoded in the next step. This can be repeated to store a potentially long sequence in the same subspace, limited by the numerical precision. We call these 'onion representations' to invoke peeling back layers corresponding to sequence positions.

Hypothesis 3 falls outside of the LRH. In linear representations, tokens are directions and each position has its own subspace. All positions are independently accessible; tokens can be read-out and manipulated given the right target subspace. Onion representations have very different characteristics.

First, tokens have the same direction regardless of which position they are stored in; the magnitude of the token embedding determines the position rather than its direction. As a result, if multiple positions contain the same token, the same direction will be added twice with different scaling factors (see figure 1d where the token $c$ occurs in positions 2 and 3). Second, because the memory is the sum of the scaled token embeddings, it is impossible to isolate the position associated with a given scale. Only the token with the most dominant scale can be extracted at a given time, by matching it to a dictionary of possible token directions. This is done by the final classifier for our GRUs. The autoregressive feedback for GRUs in effect peels off each layer, clearing access to the next variable.

Appendix F provides a toy implementation of the onion solution to elucidate the underlying concepts.

## 7.1 Intervention Data

For the causal analysis of onion representations, we do not use interchange interventions. Instead, we learn an embedding matrix for each token that

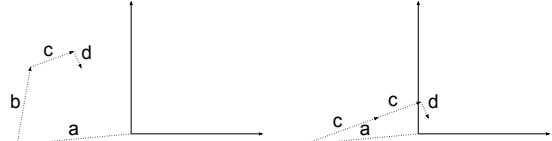

Figure 3: The intervention described by Equations 10–13 where the input sequence is $(a, b, c, d)$ and the intervention is to fix the second position to be the token $c$.

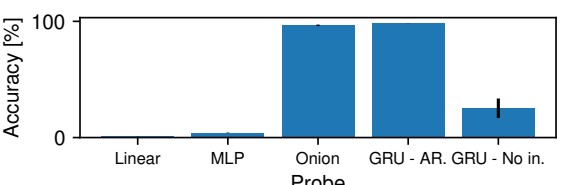

Figure 4: Accuracy of different probes on the final representation $\boldsymbol{h}_L$ of GRUs with $N = 64$ and autoregressive input (mean of 5 runs; $\pm$ 1 s.d.). Only the probes that use autoregressive denoising can successfully decode the sequence.

encodes how the model represents that token in its hidden state vector. To replace a token in a sequence $i_1 \ldots i_L$, we add the difference of the embeddings for a new $\hat{i}_j$ and old $i_j$ token scaled according to the target position $j$. Our goal is to intervene upon the hidden representation $\hat{\boldsymbol{h}}_L$ so that the sequence decoded is $i_1 \ldots \hat{i}_j \ldots i_L$. We randomly sample $\hat{i}_j$ and use inputs from the GRU training data.

## 7.2 Method: Onion Interventions

To replace token $i_j$ with token $\hat{i}_j$, we add the difference of the corresponding token embeddings scaled by a factor determined by the position $j$. We parameterize this as:

$$\boldsymbol{x} = \boldsymbol{E}[i_j] \tag{10}$$

$$\hat{\boldsymbol{x}} = \boldsymbol{E}[\hat{i}_j] \tag{11}$$

$$\boldsymbol{s} = \boldsymbol{g}\boldsymbol{\gamma}^j + \boldsymbol{\beta}j + \boldsymbol{b} \tag{12}$$

$$\boldsymbol{h}' = \hat{\boldsymbol{h}} + \boldsymbol{s} \odot (\hat{\boldsymbol{x}} - \boldsymbol{x}) \tag{13}$$

where $\boldsymbol{E} \in \mathbb{R}^{N_S \times N}$ is the embedding for the tokens (distinct from the the GRU input embedding, learned from scratch for the intervention), and $\boldsymbol{g}, \boldsymbol{\gamma}, \boldsymbol{\beta}, \boldsymbol{b} \in \mathbb{R}^N$ are learned scaling parameters. Intuitively, $\boldsymbol{s}$ is the scale used for the token in position $j$. Its main component is the exponential term $\boldsymbol{\gamma}$. In order to replace the token in the sequence, compute the difference of their embeddings, and scale them to the scale corresponding to the given position. Different channels in the state $\boldsymbol{h} \in \mathbb{R}^N$ might have different scales. Figure 3 depicts an example intervention, extending figure 1.

## 7.3 Results

The last row of Table 2 shows that our onion intervention achieves significantly better accuracy on the small models compared to the alternative unigram and bigram interventions. For example, for $N = 64$, the onion intervention achieves $87\%$ accuracy compared to the $1\%$ of the bigram intervention. As a control, if we fix $\boldsymbol{\gamma} = 1$ and $\boldsymbol{\beta} = 1$, we only reach $21\%$ accuracy.

## 7.4 Discussion

**Why do GRUs learn onion representations?** In order to distinguish $N_S$ tokens stored in $L_{\max}$ possible positions, the model needs to be able to distinguish between $N_S \times L_{\max}$ different directions in the feature space. In our experiments this is 300 possible directions, stored in a 64-dimensional vector space. In contrast, for onion representations, they only have to distinguish between $N_S = 30$ directions at different orders of magnitude.

**Onion representations require unpeeling via autoregression.** We train a variety of probes to decode the final representation $\boldsymbol{h}_L$ after encoding the input sequence of GRUs with $N = 64$, which learn onion representation. We show our results in figure 4. The *linear* and *MLP* probes predict the entire sequence at once by mapping the hidden vector $\boldsymbol{h}_L \in \mathbb{R}^N$ to the logits for each timestep $\boldsymbol{y}_{\text{all}} \in \mathbb{R}^{N_S \times L_{\max}}$. The *GRU Autoregressive (GRU – AR)* probe is equivalent to the original model, and we use it as a check to verify that the decoding is easy to learn. The *GRU – No input* probe is similar, but unlike the original decoder of the model, it does not receive an autoregressive input.

The probe results confirm that it's not merely a free choice whether the decoder uses an autoregressive input or not: if an onion representation is learned during the training phase, it is impossible to decode it with a non-autoregressive decoder, contrary to the same-size models that are trained without an autoregressive input, shown in Table 4 in Appendix E.3. We also show the special probe we designed for onion representations in a similar spirit to the intervention described in section 7.2, which performs almost perfectly. More details can be found in Appendix E.3.

**What is the feature space of an onion representation?** Together, the embeddings $\boldsymbol{E}$ learned for each token and the probe $\mathcal{P}$ that predicts the to-

ken sequence form an encoder $\mathcal{F}$ that projects the hidden vector $\boldsymbol{h}_L$ into a new feature space:

$$\mathcal{F}(\boldsymbol{h}_L) = \langle \boldsymbol{E}[\mathcal{P}(\boldsymbol{h}_L)_1], \ldots,$$

$$\boldsymbol{E}[\mathcal{P}(\boldsymbol{h}_L)_L], \boldsymbol{h}_L - \sum_{j=2}^{L} \boldsymbol{E}[\mathcal{P}(\boldsymbol{h}_L)_j] \cdot \boldsymbol{s}_j \rangle$$

where the first $L$ features are the token embeddings corresponding to the token sequence predicted by the probe and the final feature is what remains of the hidden state after those embeddings are removed. The inverse is a simple weighted sum:

$$\mathcal{F}^{-1}(\mathbf{f}) = \mathbf{f}_{L+1} + \sum_{j=1}^{L} \mathbf{f}_j \cdot \boldsymbol{s}_j$$

If the probe had perfect accuracy, this inverse would be perfect. Since our probe has 98% accuracy, there is a reconstruction loss when applying the featurizer and inverse featurizer (similar to sparse autoencoders, e.g., Bricken et al. 2023; Huben et al. 2024).

This onion feature space is parameterized by an embedding for each token, a dynamic scaling factor, and a probe. In contrast, a single linear feature is just a vector. However, because $\mathcal{F}$ is (approximately) bijective, we know that $\mathcal{F}$ (approximately) induces an intervention algebra (Geiger et al., 2024a) where each feature is modular and can be intervened upon separately from other features. **Our embedding-based interventions are equivalent to onion interchange interventions.** We evaluated the linear representations of large networks with interchange interventions that fixed a linear subspace to the value it would have taken on if a different token sequence were input to the model. There is a corresponding interchange intervention for onion representations. However, it turns out that these onion interchange interventions are equivalent to the scaled difference of embeddings used in our experiments (see Appendix B). **Why do Onion interventions also work on large models?** Surprisingly, the onion intervention works well on the big models that have linear representations of position ($N \geq 256$). We hypothesize that this is possible because all of the models start with gates open before closing them in a monotonic, sequential manner as the input sequence is processed. This enables the scaling-based onion intervention to simulate the actual gating pattern sufficiently closely to be able to perform the intervention well enough. The intervention cannot express arbitrarily sharp gate transitions but can compensate for them by creating an ensemble with different decay factors for the different channels.

From Table 5 in the Appendix, it can be seen that the onion intervention achieves significantly worse performance on the small non-autoregressive models that use linear representations compared to the autoregressive ones. This is expected, as the onion intervention cannot express an arbitrary gating pattern that might be learned by these models.

## 8 Discussion and Conclusion

The preceding experiments show that GRUs learn highly structured and systematic solutions to the repeat task. It should not be overlooked that two of these solutions (those based in unigram and bigram subspaces) are consistent with the general guiding intuitions behind the LRH and so help to illustrate the value of testing hypotheses in that space. However, our primary goal is to highlight the onion solution, as it falls outside the LRH.

Our hope is that this spurs researchers working on mechanistic interpretability to consider a wider range of techniques. The field is rapidly converging around methods that can only find solutions consistent with the LRH, as we briefly reviewed in section 2. In this context, counterexamples to the LRH have significant empirical and theoretical value, as Olah (2024) makes clear:

> But if representations are not mathematically linear in the sense described above [in a definition of the LRH], it's back to the drawing board – a huge number of questions like "how should we think about weights?" are reopened.

Our counterexample is on a small network, but our task is also very simple. Very large networks solving very complex tasks may also find solutions that fall outside of the LRH.

There is also a methodological lesson behind our counterexample to the LRH. Much interpretability work is guided by concerns related to AI safety. The reasoning here is that we need to deeply understand models if we are going to be able to certify them as safe and robust, and detect unsafe mechanisms and behaviors before they cause real harm. Given such goals, it is essential that we analyze these models in an unbiased and open-minded way.

## 9 Limitations

**The generality of onion representations.** Onion representations are well fit for memorizing a sequence in order or in reverse order, but they cannot provide a general storage mechanism with arbitrary access patterns. It is unclear if such representations are useful in models trained on more complex real-world tasks.

**Using GRU models.** Our exploration is limited to GRU models, which themselves might have less interest in the current Transformer-dominated state of the field. However, we suspect that the same representations are beneficial for other gated RNNs as well, such as LSTMs. Although we have a reason to believe that such representations can emerge in Transformers and state space models as well, we do not verify this hypothesis empirically.

**Onion representations only emerge in small models.** This might indicate that onion representations are not a problem for bigger models used in practice. However, this might not be the case: LLMs, which are much bigger, operate on an enormous feature space using a relatively small residual stream. Thus, the pressure to compress representations and the potential for similar representations to emerge could be well motivated there as well.

**Numerical precision.** The number of elements that can be stored in onion representations depends on the numerical precision of the data type used for the activations. We found that the network finds it easy to use these representations even with 16-bit floating point precision (bf16), potentially because multiple redundant channels of the state can be used as an ensemble. It remains unclear what the capacity of such representations is.

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

| Variant | $N = 48$ | $N = 64$ | $N = 128$ | $N = 256$ | $N = 512$ | $N = 1024$ |
|---|---|---|---|---|---|---|
| Autoregressive | $0.95 \pm 0.01$ | $0.97 \pm 0.00$ | $1.00 \pm 0.00$ | $1.00 \pm 0.00$ | $1.00 \pm 0.00$ | $1.00 \pm 0.00$ |
| No input | $0.88 \pm 0.11$ | $1.00 \pm 0.00$ | $1.00 \pm 0.00$ | $1.00 \pm 0.00$ | $1.00 \pm 0.00$ | $1.00 \pm 0.00$ |

Table 3: Exact-match accuracy (mean of 5 runs; $\pm$ 1 s.d.) for GRUs of different sizes trained on the repeat task results, with and without autoregressive input during the decoding.

## Appendix

## A  Performance of the Non-Autoregressive GRUs

We show the performance of all our models in Table 3, both autoregressive and those that do not receive autoregressive feedback during the decoding phase. All models solve the task well, except the smallest $N = 48$ model without autoregressive decoding. The model finds it hard to distinguish between $N_S \times L_{\max} = 300$ different directions in the 48-dimensional space. On the other hand, onion representations learned with autoregressive decoding work well even in these small models.

## B  Onion Interchange Interventions

For position $j$ and input token sequences $a_1, \ldots, a_L$ and $b_1, \ldots, b_M$, define the onion interchange intervention to be

$$\mathbf{f}^a = \mathcal{F}(\boldsymbol{h}^a)$$

$$\mathbf{f}^b = \mathcal{F}(\boldsymbol{h}^b)$$

$$\hat{\boldsymbol{h}}^a = \mathcal{F}^{-1}(\mathbf{f}^a_1, \ldots, \mathbf{f}^b_j, \ldots \mathbf{f}^a_L, \mathbf{f}^a_{L+1})$$

However, observe that that is simply the intervention of adding in the difference of the embeddings $b_j$ and $a_j$ scaled according to the position $j$ from Equations 10–13:

$$\hat{\boldsymbol{h}}^a = \mathcal{F}^{-1}(\mathbf{f}^a_1, \ldots, \mathbf{f}^b_j, \ldots \mathbf{f}^a_L, \mathbf{f}^a_{L+1})$$

$$= \mathcal{F}^{-1}(\boldsymbol{E}[a_1], \ldots, \boldsymbol{E}[b_j], \ldots \boldsymbol{E}[a_L], \mathbf{f}^a_{L+1})$$

$$= \mathbf{f}^a_{L+1} + \sum_{k=1}^{L} \boldsymbol{s}_k \cdot \boldsymbol{E}[a_k] + (\boldsymbol{E}[b_j] - \boldsymbol{E}[a_j]) \cdot \boldsymbol{s}_j$$

$$= \boldsymbol{h}^a + (\boldsymbol{E}[b_j] - \boldsymbol{E}[a_j]) \cdot \boldsymbol{s}_j$$

This means the success of our intervention $\hat{\boldsymbol{h}}$ to replace the token in $a_1, \ldots, a_L$ at position $j$ with a new token $t$ entails the success of any onion interchange interventions where we patch from an input sequence $b_1, \ldots, b_M$ with $b_j = t$. The learned token embeddings for onion representations creates a semantics for tokens that is externtal to the underlying model, so interchange interventions on the feature space have to do with the token embeddings rather than the representations actually created on the given source input. This is not the case for linear interchange interventions, where the value of the subspace intervention that must be performed is computed directly from the hidden representation created for the second input token sequence.

## C  Probe Accuracy For All Models

We show the accuracy of all of our probes in all models that we trained in Table 4. Linear and MLP probes work well when the learned solution respects LRH. Onion probes work well even for our smallest autoregressive models. We can see that autoregressive GRU can successfully decode all sequences, as expected, proving that relearning the decoding phase is a relatively easy learning problem. However, non-autoregressive GRUs are unable to decode sequences from onion representations. For more details, refer to sections 5–7.

| Decoder | Variant | $N = 48$ | $N = 64$ | $N = 128$ | $N = 256$ | $N = 512$ | $N = 1024$ |
|---|---|---|---|---|---|---|---|
| Linear | Autoregressive | $0.01 \pm 0.00$ | $0.01 \pm 0.00$ | $0.31 \pm 0.03$ | $0.89 \pm 0.03$ | $0.97 \pm 0.00$ | $0.99 \pm 0.01$ |
| | No input | $0.31 \pm 0.10$ | $0.89 \pm 0.05$ | $0.98 \pm 0.02$ | $1.00 \pm 0.00$ | $1.00 \pm 0.00$ | $1.00 \pm 0.00$ |
| MLP | Autoregressive | $0.02 \pm 0.00$ | $0.04 \pm 0.00$ | $0.55 \pm 0.04$ | $0.98 \pm 0.00$ | $1.00 \pm 0.00$ | $1.00 \pm 0.00$ |
| | No input | $0.53 \pm 0.25$ | $0.95 \pm 0.04$ | $1.00 \pm 0.00$ | $1.00 \pm 0.00$ | $1.00 \pm 0.00$ | $1.00 \pm 0.00$ |
| Onion | Autoregressive | $0.92 \pm 0.02$ | $0.97 \pm 0.01$ | $1.00 \pm 0.00$ | $1.00 \pm 0.00$ | $1.00 \pm 0.00$ | $1.00 \pm 0.00$ |
| | No input | $0.76 \pm 0.08$ | $0.96 \pm 0.01$ | $1.00 \pm 0.00$ | $1.00 \pm 0.00$ | $1.00 \pm 0.00$ | $1.00 \pm 0.00$ |
| GRU - autoregressive | Autoregressive | $0.97 \pm 0.01$ | $0.98 \pm 0.00$ | $1.00 \pm 0.00$ | $1.00 \pm 0.00$ | $1.00 \pm 0.00$ | $1.00 \pm 0.00$ |
| | No input | $0.92 \pm 0.02$ | $1.00 \pm 0.00$ | $1.00 \pm 0.00$ | $1.00 \pm 0.00$ | $1.00 \pm 0.00$ | $1.00 \pm 0.00$ |
| GRU - no input | Autoregressive | $0.10 \pm 0.02$ | $0.25 \pm 0.08$ | $0.86 \pm 0.01$ | $0.99 \pm 0.00$ | $1.00 \pm 0.00$ | $1.00 \pm 0.00$ |
| | No input | $0.77 \pm 0.07$ | $0.98 \pm 0.01$ | $1.00 \pm 0.00$ | $1.00 \pm 0.00$ | $1.00 \pm 0.00$ | $1.00 \pm 0.00$ |

Table 4: Probe accuracy (mean of 5 runs; $\pm$ 1 s.d.).

| Intervention | $N = 48$ | $N = 64$ | $N = 128$ | $N = 256$ | $N = 512$ | $N = 1024$ |
|---|---|---|---|---|---|---|
| Linear Unigram | $0.06 \pm 0.07$ | $0.37 \pm 0.17$ | $1.00 \pm 0.00$ | $1.00 \pm 0.00$ | $1.00 \pm 0.00$ | $1.00 \pm 0.01$ |
| Linear Bigram | $0.18 \pm 0.04$ | $0.95 \pm 0.06$ | $1.00 \pm 0.00$ | $1.00 \pm 0.00$ | $1.00 \pm 0.00$ | $1.00 \pm 0.00$ |
| Onion Unigram | $0.24 \pm 0.02$ | $0.41 \pm 0.04$ | $0.76 \pm 0.01$ | $0.92 \pm 0.01$ | $0.96 \pm 0.01$ | $0.98 \pm 0.00$ |

Table 5: Intervention accuracy for GRUs without an autoregressive input in the decoding phase, with different sizes, trained on the repeat task (mean of 5 runs; $\pm$ 1 s.d.).

## D GRU Models Without Autoregressive Decoding

In principle, RNN models do not need an autoregressive feedback loop during the decoding phase to be able to produce a consistent output. Given that we found that the network often relies on storing bigrams (section 6) or on onion representations (section 7), both of which benefit from autoregressive feedback, we asked what representation the models learn without such a mechanism. Thus, we changed our GRU model to receive only special PAD tokens during the decoding phase. We show the intervention accuracies in Table 5. We can see that the model is heavily based on storing unigrams, and the intervention now works down to $N = 1024$. For the $N = 64$ case, the models store bigrams. No intervention works well for the $N = 48$ non-autoregressive model, but that model also does not perform well on the validation set (see Table 3). The model is unable to to learn onion representation at any scale, since the autoregressive input is required for that, as shown in figure 4. This experiment also confirms that our subspace intervention method introduced in section 5.2 works well even for models with $N = 64$.

## E Additional Discussion of the Bigram Interventions

### E.1 Successful Unigram Interventions Entail Successful Bigram Interventions

With bigram interventions, in addition to copying a token to the randomized sequence, we also copy its neighborhood and replace two variables. In contrast, unigram interventions only move the corrupted token and replace its corresponding variable. Thus, the unigram intervention performs a subset of movements performed by the bigram. This means that if the unigram intervention is successful, it is guaranteed that the bigram intervention will be successful as well.

### E.2 Verifying the Expressivity of the Subspace Intervention

Obtaining negative results for the unigram intervention on smaller models ($N < 512$) might raise the question of whether our intervention is expressive enough to capture the relatively small subspaces of these models. In order to verify this, we trained a GRU model without autoregressive input (Appendix D) during the decoding phase. By doing this, we eliminate some of the advantages provided by bigram representations. Since GRUs are RNNs, they can learn a decoding state machine without relying on seeing the output generated so far. We confirm this in Table 3. In these modified networks, unigram interventions are successful down to $N = 128$, and the bigram intervention is successful on all scales. We show the

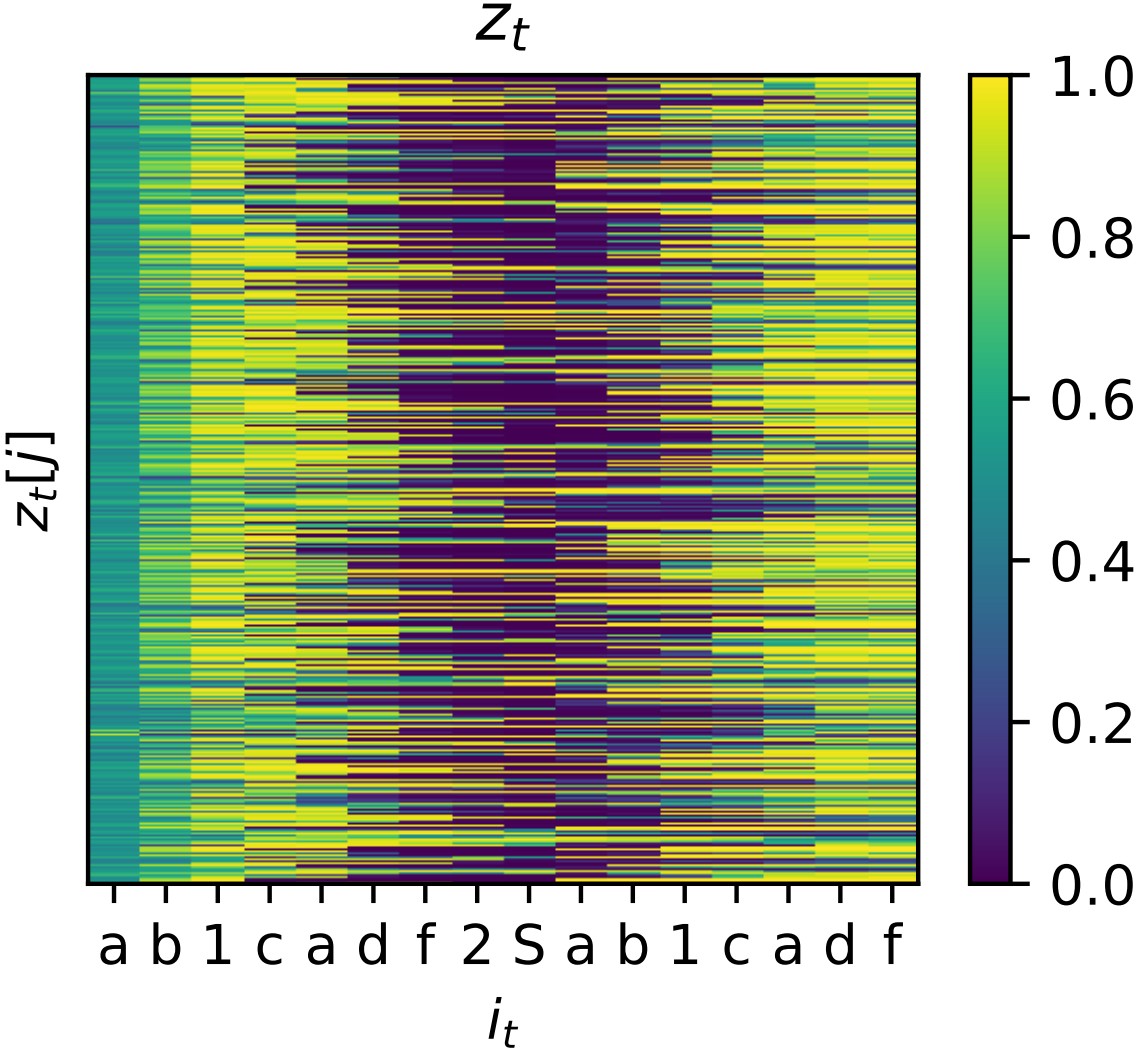

Figure 5: All 1024 channels of the GRU gate $z_t$ shown in Figure 2a. All channels follow similar patterns.

detailed results in Table 5.

### E.3 The Onion-probe

We designed a probe for onion representations similarly to the intervention described in section 7.2. We
take the final representation after encoding the sequence, $\boldsymbol{h}_L$, and decode $y_L + 1 = i_1 \ldots y_{2L} = i_L$ from
it as follows:

$$\boldsymbol{s}_t = \boldsymbol{g}\boldsymbol{\gamma}^{t-L} + \boldsymbol{\beta}(t - L) + \boldsymbol{b} \tag{14}$$

$$y_t = \operatorname{argmax} g(\boldsymbol{h}_{t-1}) \tag{15}$$

$$\boldsymbol{h}_t = \boldsymbol{h}_{t-1} - s_t \boldsymbol{E}[y_t] \tag{16}$$

As a denoising classifier $g(\boldsymbol{h})$ we use a 2 layer MLP with a layernorm (Ba et al., 2016) on its inputs
$g(\boldsymbol{h}) = \operatorname{softmax}\left(\boldsymbol{W}_{o_2} \max(0, \operatorname{LN}(\boldsymbol{h}\boldsymbol{W}_{o_1} + \boldsymbol{b}_{o_1})) + \boldsymbol{b}_{o_2}\right)$, where $\operatorname{LN}(\cdot)$ is the layernorm. Layernorm is
not strictly necessary, but it greatly accelerates the learning of the probe, so we decided to keep it.

## F Toy Model Implementing Onion Representations

To show more clearly how a model can learn to represent sequence elements in different scales, we
constructed a toy model that uses prototypical onion representations:

$$s_t = \begin{cases} 1, & \text{if } t = 1 \\ -1, & \text{if } t = L + 1 \\ \gamma s_{t-1} & \text{otherwise} \end{cases} \tag{17}$$

$$\boldsymbol{h}_1 = 0 \tag{18}$$

$$\boldsymbol{h}_{t+1} = \boldsymbol{h}_t + s_t \boldsymbol{x}_t \tag{19}$$

$$\boldsymbol{y}_t = \operatorname{softmax}\left(\boldsymbol{h}_t \boldsymbol{W}_o + \boldsymbol{b}_o\right) \tag{20}$$

where $s_t \in \mathbb{R}$ is a scalar state representing the current scale, $\gamma \in \mathbb{R}$ represents the difference in the scales
used for different variables, and $\boldsymbol{h}_t \in \mathbb{R}^N$ is the vector memory. In a real RNN, both the vector memory
and the current scale are part of a single state vector. In our experiments, we use a fixed $\gamma = 0.4$. The
inputs are embedded in the same way as for our GRU model: $\boldsymbol{x}_t = \boldsymbol{E}[i_t]$, where $i_t \in \mathbb{N}$ is the input
token and $\boldsymbol{E} \in \mathbb{R}^{N_S \times N}$ is the embedding matrix. The only learnable parameters of this model are the
embedding matrix, $\boldsymbol{E}$ and the parameters of the output projection, $\boldsymbol{W}_o \in \mathbb{R}^{N \times N}$ and $\boldsymbol{b}_o \in \mathbb{R}^N$.

The idea behind this model is based on the fact that a linear layer followed by a softmax operation
is able to 'denoise' the representation $\boldsymbol{h}_t$. $\gamma$ is chosen as $< 0.5$, because in that case the contribution to
the hidden state $\boldsymbol{h}_t$ of all future $t' > t$ positions will be lower than the contribution of input $\boldsymbol{x}_t$. Thus,
$\boldsymbol{x}_t$ will dominate all $\boldsymbol{h}_{t'}$ for all $t' > t$. Thus, when decoding from $\boldsymbol{h}_{t'}$, Eq. 20, followed by the argmax
used in greedy decoding, the model will always recover the first, most dominant $i_t$ that is not yet decoded
from the model. Then, this token is autoregressively fed back to the next step, where it is subtracted from
$\boldsymbol{h}_{t'}$, letting the next token dominate the representation $\boldsymbol{h}_{t'+1}$. This allows storing an arbitrary sequence
at different scales of the representation $\boldsymbol{h}_t$. All 5 seeds of this model that we trained achieve perfect
validation accuracy.

