# OpenReview forum: "Recurrent Neural Networks Learn to Store and Generate Sequences using Non-Linear Representations"
_EMNLP/2024/Workshop/BlackBoxNLP — BlackboxNLP 2024_

### Official Review · Reviewer_ZDpu · 2024-09-06

**Overall Assessment:** 4
**Confidence:** 3

**Best Paper:**

2

**Best Paper Justification:**

The paper challenges the widely accepted linear representation hypothesis by offering compelling evidence that smaller RNN models can learn non-linear representations.

**Comments Questions Suggestions And Typos:**

- In line with the unigram and bigram hypotheses, it would have been interesting to test whether the behaviour of the smallest models could be explained by a trigram hypothesis. If this was investigated or if there is a valid reason why it cannot be applied in this context, it would be helpful to clarify this in the Appendix.
- The title seems to be overclaiming the findings, as the paper only provides evidence that the smallest models adopt non-linear strategies, while larger models still rely on linear representations. A more accurate phrasing might be “*Can* Learn to Store and Generate Sequences using Non-Linear Representations,” which better reflects the nuanced results.

**Paper Summary:**

The paper presents a counterexample to the linear representation hypothesis, a common assumption in the field of interpretability. The authors study the synthetic task of repeating the input sequence consisting of random tokens sampled with replacement, and train a suite of RNNs with GRUs with varying hidden sizes ranging from 48 to 1024. Then, they investigate how the model solves the task for each of these models, testing different hypotheses using DAS. They find that the largest models (512, 1024) solve the task by representing each combination of token and position linearly (unigram hypothesis). Smaller models (128, 256) appear to solve the task by linearly storing tuples of input (bigram hypothesis). Each of these hypotheses are verified using causal interventions. But the authors find that none of these hypotheses explain the behaviour of the smallest models (48, 64). Thus, they investigate whether it might be representing the information non-linearly. Specifically, the authors hypothesise that the model uses an onion representation, where multiple features can be stored in the same subspace at different scales. They design a custom intervention that involves learning an embedding matrix for each token in the hidden state vector and scaling this embedding according to the target position. Surprisingly, the authors find that this intervention explains most of the model behaviour for the smallest models, suggesting that it uses non-linear representations to solve the task.

**Summary Of Strengths:**

- The study is well motivated and of potentially high impact, as a significant number of methods rely on the assumption that features are encoded linearly (e.g. sparse autoencoders).
- I enjoyed reading the paper, as it guides the reader through the thought processes that led to the interesting finding. I also enjoyed the thought-provoking discussion and conclusion section.

**Summary Of Weaknesses:**

- The onion unigram intervention also achieves high accuracy in larger models (Table 2). The authors hypothesize that this is due to the scaling-based onion intervention mimicking the gating patterns of GRUs. To address this, they evaluate the interventions on a non-autoregressive model, where the onion intervention shows significantly worse performance in smaller models. However, it still raises concerns about whether the intervention might be too powerful and subsume other interventions that would provide a linear explanation for the behaviour of the smallest models.

---

### Official Review · Reviewer_VWLQ · 2024-09-08

**Overall Assessment:** 2
**Confidence:** 4

**Best Paper:**

1

**Best Paper Justification:**

not applicable

**Comments Questions Suggestions And Typos:**

- Equations of the GRU lines 176-178: it should be h_{t-1} instead of h_t.

- Lines 358-363: this is very unclear to me

**Paper Summary:**

This work aims to demonstrate that the strong version of the "Linear Representation Hypothesis (LRH)" does not hold for recurrent networks by showcasing a counterexample toy task where this hypothesis fails. The considered toy task is simply sequence copying, and the considered model is a standard GRU (one layer and unidirectional I suppose, as no details are given on the model other than the hidden layer size).

The LRH states that neural networks learn to encode concepts as directions in activation space (or in other words, that concepts are linearly encoded in activation space, thus they can be recovered by linearly projecting the model's activations).

In order to falsify this hypothesis, the authors show on their considered toy task (i.e. sequence copying), that IDENTICAL tokens AT DIFFERENT TIME POSITIONS in the sequence are NOT encoded in SEPARATE directions, but in the SAME direction and that the distinction on the tokens' position is then made on the MAGNITUDE of vectors in that token's direction. The authors call this type of encoding "onion-based representation".

In particular the authors show (through an intervention analysis, i.e. by modifying the GRU's hidden state on purpose to replace a token by another) that the onion representation holds for small GRUs (hidden size up to 64 units). For larger GRUS the authors find the tokens' positions are encoded in different linear subspaces, though these linear representations are still compatible with their proposed onion-based representation (which sounds a bit contradictory to me, and would mean that the larger models have both a linear subspace as well as an onion-based representation).

**Summary Of Strengths:**

- A timely topic: the Linear Representation Hypothesis

- Proper Analysis of the Information stored in the Hidden Vector of a GRU based on Interventions

**Summary Of Weaknesses:**

First of all, I think this paper would benefit from clarifying from the beginning various terms as well as the general motivation. What are "concepts" in LRH in general and in this work in particular, how do the authors understand the LRH itself, and what they want to show through their toy task.

For instance, it is only when I read their work a second time that I understood that what the authors call "concepts" in their sequence copying task are not the token's identity (i.e. distinguishing between token A and token B in the vocabulary), but rather their TIME POSITION within the sequence (distinguishing between token A at position {t} and the same token A at position {t+n}). In fact for the considered toy task of copying a sequence of tokens, there can only be two concepts involved: "which token" shall be emitted and "when" shall it be emitted. Although this seems obvious after thinking a little bit about it, the authors shall state this clearly to the reader.

Further, it seems the authors focus primarily on the second concept: "when" a token shall be emitted, or in other words how the token's position is encoded. Why do they focus only on this concept?

Now concerning the LRH, it is not obvious to me that the strong version of this hypothesis assumes that concepts are encoded in SEPARATE linear subspaces for each concept.
In my understanding the LRH only states that concepts are linearly encoded, which means with simple linear algebra (linear projection) you can identify the concepts, AS OPPOSED to encoding them NON-LINEARLY. Which means that you can recover concepts for instance through linear probing as opposed to requiring a non-linear function to unveil the concepts. So I don't think having a concept encoded in a single direction and on different scales in this single direction is contradicting the LRH (I mean it is still a linear encoding then and not a non-linear one).

Now where there is room for interpretation I believe in the LRH is what concepts you are looking for (somehow this shall be a "subtask" of your whole training task), and which representations you look at where these concepts shall be stored (which hidden layer vectors or embeddings you look at where the concepts can be stored).

In your task you take directly the output vector of the entire neural network (i.e. the last time step's GRU hidden state vector), which during training and inference is then fed to a final LINEAR output layer that makes the prediction of the next token to emit. Because for solving the task you enforce the GRU hidden state to encode the problem in a linear manner (I mean the hidden state vector is followed only by a linear layer), it seems quite obvious to me that then the information you need for prediction (which token to emit) is then contained in the hidden state in a linear manner (as the linear output layer on top of it solves the task using the hidden state vector as input). So I'm quite unsure what is the purpose of testing the LRH on the hidden state vector (rather than maybe any gate or other hidden vector within the GRU). Now your goal could be to unveil whether the concept "time position" is encoded in "separate" linear subspaces, or if it is stored in the "same" subspace, the latter variant then inevitably would make the scale in that direction matter. But I don't see how this is related to the LRH.

So while I understand somehow what you are trying to show at the low-level (analyzing different ways a vector can store information linearly, i.e direction and/or scale), I am unsure that this contradicts or proves the LRH is not true, or what impact this distinction may have on future neural network analyzes.

Further, it would be interesting to extend your approach of contradicting the LRH through a counterexample to other architectures as well. For instance in transformers, which have an explicit encoding for the time position, you would probably need to consider another concept to test.

Some more fine-grained remarks, as well as questions to the authors:

- page 2 last §: You write "RNNs fell out of favor somewhat". Please be more precise here why RNNs were replaced by other models. Later you write "We chose GRUs for our studies, with an eye towards better understanding structured state space models as well.", this is a little overstated as I don't see how the present work contributes to understanding state space models.

- Lines 236-237 you write: "Intuitively, to solve the repeat task, the token at each position will have a different feature in the state vector h_L". Since you are generating one token at a time with h_t, in my understanding you only require h_t to contain the information of the next token to generate and not of the entire sequence (this would have been different if you would have used a sequence-to-sequence model with two GRUs to solve the task instead, but you are not, you are using an encoder-only single GRU model)

- Lines 280-286 you write: "the first input of a repeated sequence, b1 , is always present, and the probability of successive inputs decreases due to the random length of the input sequences. Thus, the network might decide to allocate a larger subspace to the more important variables that are always present, maximizing the probability of correct decoding for popular sequence elements." I believe all sequence elements are equally important in the loss you are using for training, and the probability of a token occurring in a position is equal for all positions, so I don't see why the model shall allocate more importance to the first tokens in the sequence.

- Figure 2b: I don't see gates z_t closing simultaneously like you claim, in fact I see the opposite, many gates having a value of 1 (i.e. they are open) during sequence generation

- I don't understand how you go from Hypothesis H1 to H2: you look at encoding of unigrams first in H1, and then of bigrams in H2, for smaller GRUs. In my understanding storing bigrams in different directions is even harder and less likely, since you then need to distinguish bigrams of tokens (and there are more combinations of it than of unigrams). Could you explain in more details the intuition for switching from H1 to H2?

---

### Official Review · Reviewer_vVkc · 2024-09-09

**Overall Assessment:** 4
**Confidence:** 3

**Best Paper:**

1

**Best Paper Justification:**

N/A

**Comments Questions Suggestions And Typos:**

The writing is very clear, so I have no further suggestions. I was not able to find any typos either.

**Paper Summary:**

This paper provides a counterexample to the strong version of the Linear Representation Hypothesis, which posits that 'meaningful features' are encoded only linearly in the representations of neural networks. Using gated recurrent unit models of different trained on the sequence repeat task, the authors show that they learn positional representations through different vector magnitudes in a non-linear "onion-like" way.

**Summary Of Strengths:**

1) The paper is very clearly written, and the logic is easy to follow.
2) The experimental setup is simple yet effective at providing evidence for the main claims.
3) The authors are thorough in discussing/verifying alternative explanations through additional analyses.

**Summary Of Weaknesses:**

My only concern is with what the authors discuss as the first limitation. I think the argument could be made even stronger if the experimental setup was close to what was used to provide support for the LRH (but of course, I think the current experiments are enough to provide strong evidence against the strong version).

---

### Decision · Program_Chairs · 2024-09-20

**Decision:**

Accept

**Comment:**

The reviewers appreciate the way the paper engages with the "Linear Representation Hypothesis", commending the presentation, experimental setup and discussion of the results. Reviewer VWLQ, however, has various concerns about the core findings of the paper, in particular about the linear nature of the "onion representation" that is found. The authors are encouraged to take these concerns into account in the camera ready version of the paper, as well as the other "fine-grained remarks" provided by this reviewer.